

# Current rapid global temperature rise linked to falling SO₂ emissions

Nick E. B. Cowern

School of Engineering, Newcastle University, Newcastle upon Tyne, NE1 7RU, UK

*Correspondence to*: nick.cowern@ncl.ac.uk

**Abstract.** It is widely held that global temperature variations on time scales of a decade or less are primarily caused by internal climate variability, with smaller contributions from changes in external climate forcing such as solar irradiance. This paper shows that observed variations in global mean surface temperature, $T_{GS}$, and ocean heat content (OHC) during the last 1–2 decades imply major changes in climate forcing during this period. In a first step, two independent methods are used to evaluate global temperature corrected for ocean–atmosphere heat exchange. El Niño/Southern Oscillation (ENSO) corrected $T_{GS}$

(written as $\bar{T}_{GS}$) is shown to agree closely with a novel temperature metric θ that combines uncorrected $T_{GS}$ with scaled OHC. This agreement rules out a substantial 21st-century contribution to $\bar{T}_{GS}$ from ocean-atmosphere heat exchange. In contrast to $T_{GS}$, the time series $\bar{T}_{GS}(t)$ provides a clear fingerprint of transient global cooling associated with major volcanic eruptions, enabling a more accurate empirical estimate of the climate response of the global mean surface. This allows more accurate estimation of the net climate forcing by stratospheric aerosols and solar irradiance, which is then subtracted from $\bar{T}_{GS}(t)$ to

determine the underlying signal of anthropogenic global warming. Key features of this signal are a slowdown from the late 1990s to 2011 – corresponding to the well known climate hiatus – and a subsequent sharp upturn indicating a steep increase in anthropogenic climate forcing. It is argued that the only plausible cause for this increase is a large fractional decrease in tropospheric aerosol cooling. This attribution is supported by satellite-based observations of a >50 % decrease in SO₂ emissions from large sources during the last six years. It suggests that current clean-air policies and replacement of coal by natural gas

are driving a significant human made climatic event, 2–4 times faster than greenhouse driven warming alone. If confirmed, this implies a considerably shortened timescale to meet the IPCC 1.5°C objective, with major implications for near-term carbon emission policies.

## 1 Introduction

In the last decade humanity has experienced an apparent increase in the rate of global warming, with accelerating rates of

surface warming, ocean warming, mass loss from polar ice-sheets, and sea level rise. There has been much speculation over the relative influences of climate variability and underlying global temperature forcing, the latter arising largely from the effects of greenhouse gases and aerosols, upon this recent acceleration. The question is of great importance, as an attribution to temperature forcing would predict a continuation of the faster warming trend well into the next decade.



This paper uses a comparison between two largely independent measures of global temperature, one potentially dependent on non-ENSO climate variability, the other independent of all internal climate variability, to show that the primary cause of the recent faster warming rate is a sharp increase in global temperature forcing during the last five to six years. The uptick is too abrupt to be explained by changes in any of the warming greenhouse gases (GHGs), as their atmospheric lifetimes are too

long. It is however uniquely consistent with a major, rapid reduction in cooling by the short lived sulfur aerosols that form as a result of anthropogenic $SO_2$ emissions. These aerosols form atmospheric haze and influence cloud formation in ways that have historically reflected away a small, but significant, part of Earth's incoming solar radiation, offsetting about 0.5°C of potential global warming (Hansen 2017). Compellingly then, it turns out that $SO_2$ is one of the few atmospheric pollutants whose emissions have fallen sharply in recent years.

In Section 2 the temperature metrics of interest for this work are presented and discussed. Based on these results, Section 3 evaluates the warming contributions from solar intensity variations and stratospheric aerosols injected by volcanic eruptions, in order to determine more accurately the larger warming contribution from anthropogenic sources. Section 4 discusses the relationship between the recent warming acceleration and falling anthropogenic $SO_2$ emissions measured by the satellite borne Ozone Monitoring Instrument over the last decade. The paper concludes with a brief discussion of projected faster warming

and alternative policy approaches to mitigating it.

In summary, during the last 5–6 years global temperature has been increasing 2–4 times faster than the historical warming rate which arose primarily from greenhouse gas emissions. The effect is an unintended consequence of the recent rapid reduction in $SO_2$ emissions driven by clean air initiatives, particularly the replacement of unmitigated coal fired power plants with clean coal and gas fired plants. Owing to the slow response tail of the climate system and likely continued reductions in $SO_2$, further

warming is in the pipeline and could potentially bring forward the date when warming reaches 1.5°C to the late 2020s, sooner than estimated by the IPCC special report on 1.5°C global warming (IPCC, 2018). This result underscores the extreme urgency with which humanity must cut emissions of short-lived greenhouse gases such as methane and ozone-generating hydrocarbons, and pursue a fast transition to a zero carbon economy.

## 2 Global temperature evolution, climate variability and climate forcing

### 2.1 ENSO corrected global mean surface temperature

Global mean surface temperature, $T_{GS}$, is a global measure reconstructed from measurements that sample air temperature just above the land surface and water temperature just below the ocean surface (Hansen et al., 2017). It is the most widely used global temperature metric, well attuned to the human experience of climate and its intrinsic variability driven by processes such as ENSO. However, for the same reason uncorrected $T_{GS}$ is a poor measure of the underlying warming produced by

climate forcing agents such as greenhouse gases and aerosols. A temperature signal that reduces intrinsic variability can be obtained by explicitly removing the ENSO signal from the $T_{GS}$ time series. Owing to the response time of the global atmosphere


to changes in sea surface temperature in the El Niño affected Pacific, the ENSO component of $T_{GS}$ is delayed by about six months with respect to the Ocean Niño Index, $T_{ONI}$ (NOAA, 2018). Here I apply the simplest available method to account for this delay, a corrected temperature

$$\bar{T}_{GS}(t) = T_{GS}(t) - \varepsilon . T_{ONI}(t - t_r).$$

Using the values $t_r = 0.5$ y and $\varepsilon = 0.1$, correlations between the $\bar{T}_{GS}$ and ENSO signals are reduced to an insignificant level and the magnitude of intrinsic variability in $\bar{T}_{GS}$ is also significantly reduced, as shown in Fig. 1. The magnitudes of the peak cooling responses to sulfur aerosols injected into the stratosphere by major volcanic eruptions in 1963, 1982 and 1991 (Textor

et al., 2003) are in the region of $0.2 - 0.3°C$, the decrease of $0.3°C$ for Mt Pinatubo being at least a factor 2 weaker than the previously reported $0.7°C$ peak response of ENSO-corrected lower troposphere temperature, $\bar{T}_{LT}$ (Soden et al., 2002). The $\bar{T}_{GS}$ time series also provides insight into more recent temperature changes, in particular, a slowdown in warming during the first decade of this century, both in the uncorrected and ENSO-corrected temperature records, and a more recent steep rise in temperature most clearly evident in the corrected record.

**2.2 Earth system temperature metric**

Up to now the most widely accepted explanation for the variations discussed has been intrinsic climate variability associated with heat transfers between the ocean and atmosphere, while decadal variations caused by changes in climate forcing have been thought to be small (Trenberth, 2015). Since the advent of accurate ocean temperature measurements by the Argo sensor network this viewpoint can be tested rigorously against experiment. Here I propose an 'Earth-system' temperature metric, $\theta$,

consisting of a weighted sum of $T_{GS}$ and the upper-ocean heat content anomaly $H_{UO}$, scaled to equivalent surface temperature. The weighting is chosen to cancel opposing variations in $T_{GS}$ and scaled $H_{UO}$ caused by heat exchange between the bulk ocean and the global surface, the thickness of upper ocean included in the metric being chosen here to be 700 m, sufficient to capture variability driven by fluctuations in ocean heat transport on decadal timescales and below (detailed analysis in Supplementary Information). The symbols in Fig. 1(b) show the result. Values are given for the period since 2002, based on analysis of

accurate ocean temperature measurements available since 2005 (Ishii and Kimoto, 2009), (JMA, 2018), (Cheng et al., 2017) and on the model-assisted analysis approach of Ishii and Kimoto (2009) applied to data of the Japan Meteorological Agency (2018) for the three previous years when available ocean data were sparser. Using the $\theta$ metric, the strong peak in $T_{GS}$ associated with the very large El Nino event in 2016 is removed. Moreover, in principle $\theta$ removes all ocean-atmosphere heat-exchange fluctuations, including any arising from the quasi-biennial oscillation (QBO) and the Pacific Decadal Oscillation (PDO) (see

Supplementary Information).



Fig. 1 (b) shows that the trend in $\overline{T}_{GS}$ during this period is consistent with the trend in θ, even though the θ-weighting takes an ≈80% contribution from scaled $H_{UO}$ and only a ≈20% contribution from $T_{GS}$ (Supplementary Information). The only substantial difference between the curves is short-term variability associated with the QBO, which shows up as fluctuations in $\overline{T}_{GS}$ but not in θ. A subtler difference arises because θ is dominated by the thermal response of the upper ocean, leading to a heavier-tailed climate response and a smoother slope change than that of $\overline{T}_{GS}$, which rises steeply after 2011.

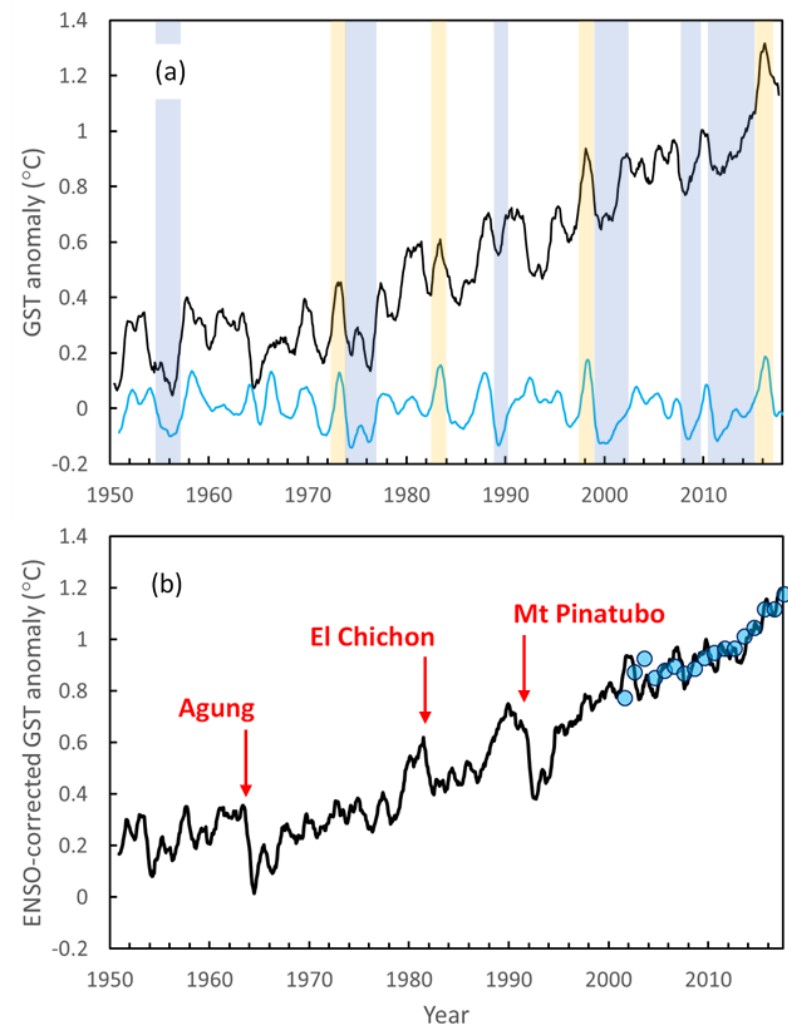

**Figure 1: (a) Monthly global mean surface temperature anomaly (black curve) and 1/10 × Ocean Nino Index delayed by 6 mo (blue curve), presented as 12-mo running means. The four 'very strong' El Niño events and six 'strong' La Niña events during this period (NOAA 2018a) are shaded in orange and blue, respectively, to illustrate the close correlation between the two time-series. (b) ENSO-corrected global mean surface temperature, showing strongly reduced fluctuations and the global cooling signatures of three major explosive volcanic eruptions, which are only weakly identifiable in the unprocessed GST signal. Symbols represent the effective temperature anomaly θ of the combined global mean surface and 0–750 m upper ocean, which in principle eliminates the perturbing effects of all ocean-atmosphere heat exchange processes, although at the cost of a more damped climate response (see text).**





However, the slope of the θ time series still nearly doubles between the periods 2005-2012 and 2012-2017. These results provide no support for a significant contribution to $\bar{T}_{GS}$ from ocean-atmosphere heat exchange linked to non-ENSO climate variability, despite a transition in 2013 from a negative to a positive phase of the PDO (Trenberth, 2015). In contrast, they

5      strongly suggest that the recent change in global warming rate is a response of the global ocean-atmosphere system to a change in climate forcing. This conclusion is consistent with previous analysis by Cheng et al. (2015) showing that climatic fluctuations in the heat content of the upper ocean averaged over the 0–750 m depth range have been primarily associated with changes in the relative frequency of El Niño and La Niña events, which are already accounted for in Fig. 1(b).

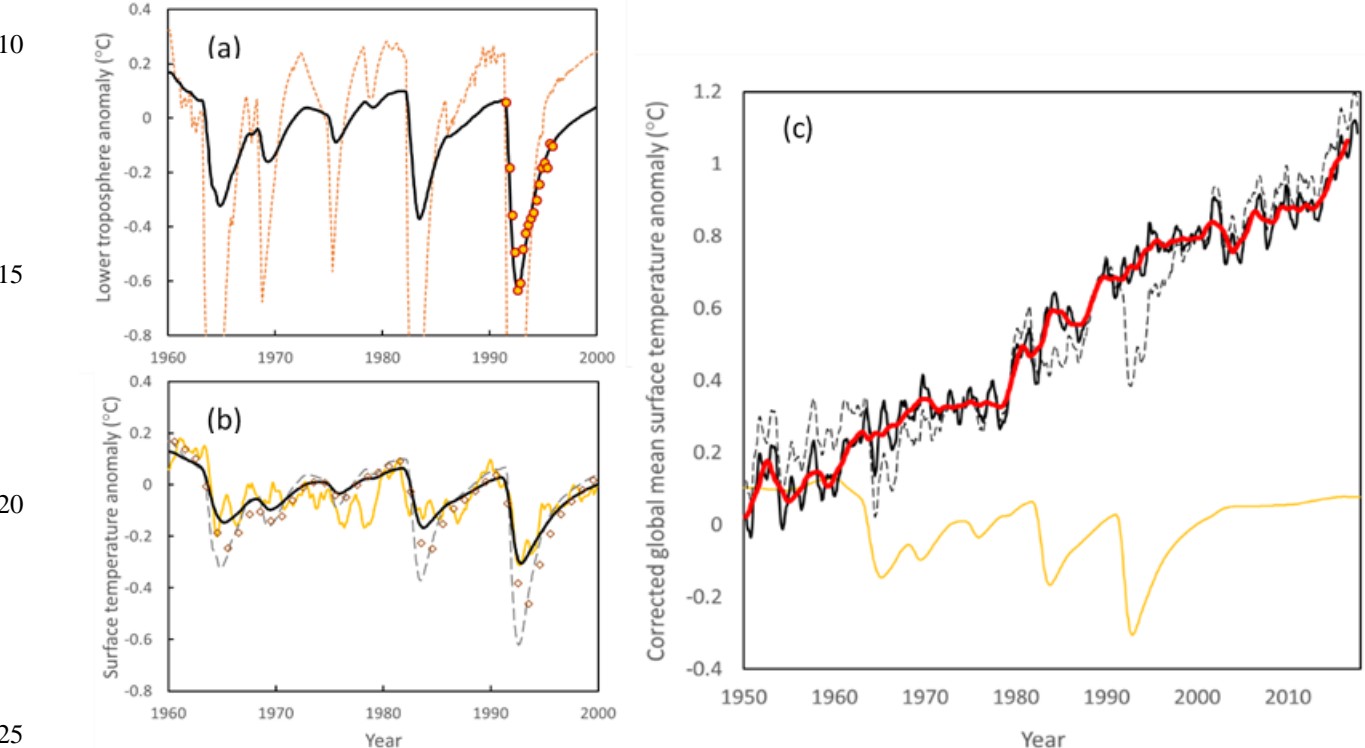

**Figure 2: (a) Climate forcing by volcanic aerosols and solar irradiance variations between 1960 and 2000 (dashed curve) and the resulting temperature evolution of the lower troposphere (solid black curve) obtained by convoluting the forcing data with the lower troposphere response function described in the text. Symbols represent satellite measurement data. (b) Global mean surface temperature evolution obtained by the same method (solid black curve) using the updated global surface response function described in the text. The dashed curve in (b) is the tropospheric response and symbols denote the time series calculated from the annual forcings and global surface response function used by Hansen et al. (2017). The gold line represents temperature anomaly data from Fig. 1(b) after subtracting a linear background representing the rising temperature trend from anthropogenic emissions. (c) Historical evolution of the ENSO-corrected global mean surface temperature anomaly (dashed line), the temperature response to volcano and solar irradiance variations (gold solid line), and the global mean surface temperature signal after removal of the volcano and solar contributions (black solid line). The red curve shows the corresponding result after suppression of quasi-biennial oscillations using a 26-month running mean. The recent increase in slope indicates a steep rise in anthropogenic global temperature forcing, confirming the results in Fig. 1.**





## 3 Volcanic eruptions, climate response, and the anthropogenic contribution to global warming

The shape of the volcano dips in Fig. 1(b) suggests that their influence on temperature change in this century is small and that the slowdown and recent rapid rise in global temperature are likely of anthropogenic origin. However, current understanding of the short-term response of global mean surface temperature to a forcing impulse $\delta(t)$ is limited, with commonly used

models typically overestimating the magnitude of the volcano dips. In order to correct this, I have revisited the previous work of Hansen et al. (2005, 2011, 2017) in order to estimate semi-empirical climate pulse-response functions, $P_{LT}$ and $P_{GS}$, for the lower troposphere and global mean surface, respectively (details in Supplementary Information). Fig. 2 shows results for the temperature responses $T(t) = \lambda \int F_e(t-t')P(t')dt$, where $F_e$ is the net forcing in W cm$^{-2}$ and $\lambda$ is the equilibrium fast climate response. Here I use $\lambda = 0.75$°C / W m$^{-2}$, a mid-range choice supported by evidence from Ref. 13 and this work (see

Supplementary Information). Owing to the influence of the ocean, which generally acts as a heat sink (Hansen 2017), $P_{GS}$ has a significantly attenuated short-term and slightly enhanced long-term response, as shown in Fig. 2 (b).

Using this updated $P_{GS}$ the contribution of volcano and solar irradiance variations, given by the solid black curve in Fig. 2(b), can be subtracted more accurately from the long-term ENSO-corrected time series in Fig. 1, in order to estimate the evolution of global mean surface temperature arising from anthropogenic climate forcing. Fig. 2 (c) shows the resulting time series. On

short time scales the series contains QBO fluctuations (Zhi-Xiu, 2013), which are not removed by the ENSO correction, but these can largely be filtered out using a 26-month moving average (thick red curve). It is clear from Fig. 2(c) that the correction for volcano and solar forcing from 2000–2017 has been small, in the range 0°C to −0.05°C, with a time variation that tends to slightly flatten the underlying warming up to 2012 and marginally reduce the rate of recent temperature rise.

The recent acceleration in global temperature rise is of great interest and concern. Fig. 3 shows detail of the results from Fig.

1(b) for the period 1995–2017.5, including the evolution of ENSO-corrected temperature, $\bar{T}_{GS}$, and of $\theta$ based on two independent analyses of ocean heat content data (JMA, 2018), (Cheng et al., 2017), (Cheng, 2018) during the ARGO period. Also shown are the uncorrected $T_{GS}$ and scaled upper-ocean heat content curves.

The magnitude of the increase in $T_{GS}$ during the 2016 El Nino event, shown by the difference between the red dashed and black solid curves, is consistent (see Supplementary Information) with the quantity of heat transferred from ocean to atmosphere,

shown by the difference between the blue dashed and black solid curves. All of the curves exhibit a long-term rising slope, whereas a decadal shift to a warmer atmosphere and cooler ocean would bend the ocean curve downwards. The average slope of the $\theta$ time series increases by a factor of two between the periods 2005-2012 and 2012-2017, and that of $\bar{T}_{GS}$ increases from nearly flat during the decade from 2002-2012 to an average slope of 0.4 – 0.5°C/decade in the period from 2012 to mid-2017. It is inescapable, then, that the recent acceleration of global temperature is the response of the Earth system to a decadal

increase in climate forcing.

Only one influence can have produced this sharp increase in climate forcing. Changes in all of the major greenhouse gases ($CO_2$, $CH_4$, $N_2O$ and CFCs) are ruled out because they or their reaction products have long or intermediate atmospheric lifetimes and are thus slowly varying atmospheric constituents. Black carbon emissions (Wang, 2014) and tropospheric ozone





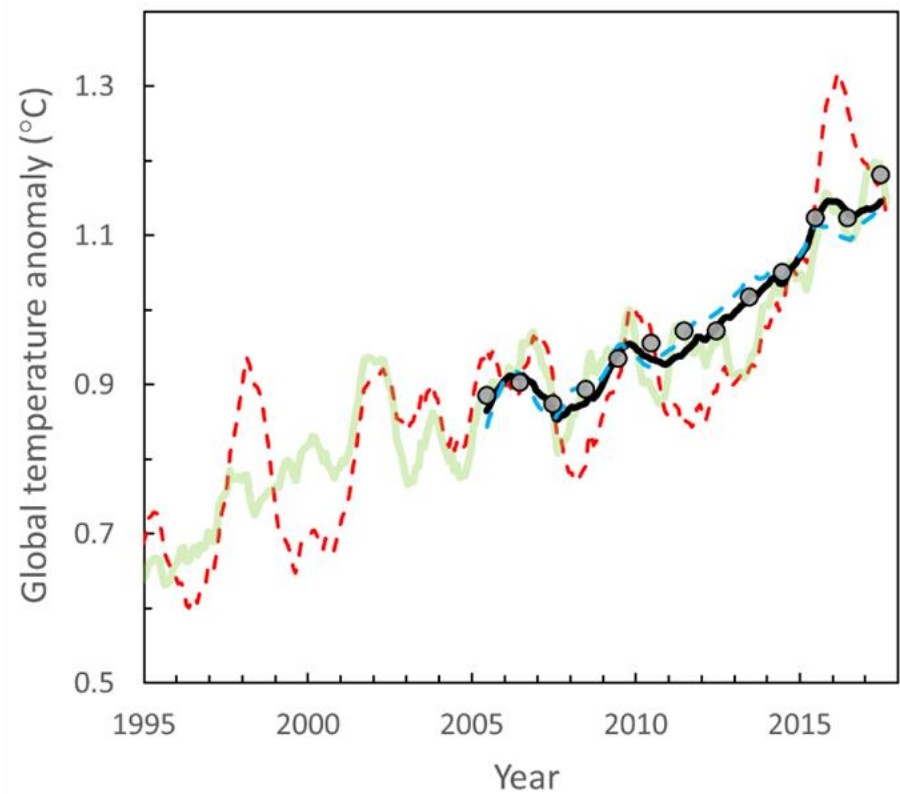

**Figure 3: Estimated global temperature rise during the period 1995–2017.5. Red dashed curve: global mean surface temperature . Blue dashed curve: scaled upper ocean heat content from Cheng et al. (2017) and Cheng (2018). Heavy black curve: $\theta$ metric, which eliminates the peak in $T_{GS}$ and the corresponding dip in upper ocean heat content. Symbols: the same $\theta$ analysis using upper-ocean heat content data from the Japan Meteorological Agency (JMA, 2018). Pale green curve: ENSO-corrected global mean surface temperature $\overline{T}_{GS}$. The $\theta$ metric eliminates most of the residual QBO signal seen in the ENSO-corrected curve. However, owing to its heavy-tailed response function, its response to forcing changes is more gradual than that of the ENSO-corrected curve. Despite this smoothing effect the average slope of $\theta$ during the period 2012-2017 is approximately double that from 2005-2012.**

concentrations (Helmig et al., 2016) may have risen as a result of increased diesel use and rising emissions of non-methane hydrocarbons from the oil and gas industry, respectively, but not sufficiently to cause a drastic increase in forcing. The sole plausible candidate is $SO_2$, which forms short-lived climate-cooling aerosols (McNeill, 2007) that have historically been a powerful counterweight to greenhouse warming (Hansen et al., 2005).

## 4 An unintended consequence: recent faster warming driven by governmental anti-pollution measures

In the last decade, efforts to reduce pollution from coal-fired power plants and other sources, together with a global trend towards replacement of coal by natural gas, have led to a strong decrease in $SO_2$ emissions. In the geographical area of China, until recently the world's largest emitter of $SO_2$, emissions from large sources measured by the Ozone Monitoring Instrument



(OMI) on EOS Aura fell by approximately 80% over the four years from 2012–2016 (Li et al., 2017), with comparable reductions in emissions from smaller sources likely as a result of domestic pollution-control measures. Moreover, global $SO_2$ emissions from (coal) power plants, currently the dominant flux of $SO_2$ to the atmosphere, fell by about 40% from 2007–2014 (Fioletov et al., 2017). The large fall in OMI-measured $SO_2$ emissions in China is especially significant because deep

5   circulation in that region can loft aerosols to the tropopause (Neely et al., 2014), (Lau et al., 2018) and lower stratosphere (Bourassa et al., 2012), where their negative climate forcing effect may be an order of magnitude stronger than in the lower troposphere. It is therefore also plausible that the strong rise in $SO_2$ emissions in East Asia in the first decade of this century contributed to the climate hiatus during that decade. Because the sulfur aerosols formed from $SO_2$ have a short atmospheric lifetime (< 2 y), the very large decrease in emissions since 2011 will inevitably have fed thro

10   aerosol forcing.

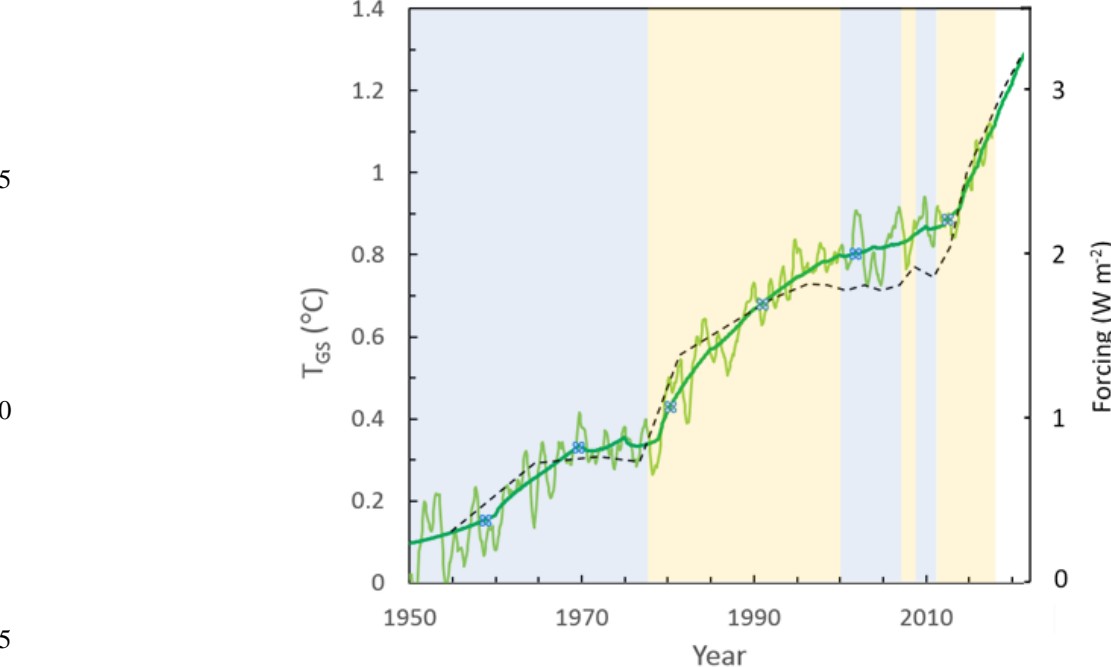

**Figure 4: Global mean surface temperature corrected for ENSO, volcano and solar irradiance contributions (light green curve) together with an illustrative fit that ignores biennial fluctuations (dark green solid curve). The black dashed curve represents the net climate forcing by anthropogenic greenhouse gases and tropospheric aerosols which is needed to generate the dark green temperature curve. The near-term projection to 2022 is based on extrapolation of the current forcing trend (see text). Blue and orange shading correspond to periods of rising and falling global SO2 emissions, respectively, based on inventory estimates from 1950–2005 (Smith, 2011), (Klimont, 2013) and satellite observations from 2005 onwards (Fioletov et al., 2016), (Li et al., 2017).**

Fig. 4 illustrates the magnitude of changes in forcing required to account for the observed evolution of $\overline{T}_{GS}$. Two plateau regions

35   in the periods 1960–1975 and 2000–2011 coincide with phases of rapid growth in $SO_2$ emissions which effectively stall the rising forcing caused by greenhouse gases. The subsequent rises correspond to periods of falling $SO_2$ emissions, which



reinforce the greenhouse warming trend. The first plateau corresponds to a 15–20 y period when global $SO_2$ emissions rose by ~2× to a record 130 Mt $y^{-1}$ in 1975 (Smith et al., 2011), which likely enhanced the negative aerosol contribution to climate forcing by nearly $-0.5$ W $m^{-2}$, while the second corresponds to a period of rapid emissions growth in China (Klimont, 2013). The recent rise, which is several times steeper than the greenhouse forcing trend, corresponds to a rate of decrease in negative

aerosol forcing of ~16% $y^{-1}$, consistent with recent $SO_2$ emissions reductions from large sources (Fioletov, 2016), (Li et al., 2017). The extrapolation to 2022 illustrates the potential for further significant near-term warming, if the recent fall in aerosol forcing and steady rise in greenhouse gas emissions are projected at the same proportional rate as in the last five years. This further rise may become less steep if emissions elsewhere, for example, from unmitigated coal-fired power plants in India (Li et al., 2017), rise in the near future prior to transition from coal to low-carbon energy.

Negative aerosol forcing in recent history has offset about 40% of net climate forcing (Hansen et al., 2017) with about 0.5°C of additional warming to be expected if this contribution is removed. Consequently, it is no surprise that a factor > 2 decrease in $SO_2$ emissions should have such a powerful warming effect. This is the payoff from Hansen's 'Faustian bargain' which has been extensively discussed (Hansen and Lacis, 1990) but has until recently been viewed as a long-term issue (Hansen, 2017), although this year subtle hints of potentially imminent changes have emerged from processes such as Arctic sea ice decline

(Mueller, 2018). In reality, that future has already arrived and is driving a significant global climatic event.

## 5 Conclusions

El Nino corrected global mean surface temperature data suggest that, since 2011, global warming has accelerated to 2–4 times the long-term warming rate. An Earth system temperature metric comprised of upper-ocean and uncorrected global mean surface temperature data confirms an acceleration in warming since 2011, indicating that the rise is not a result of variability

in ocean–atmosphere heat exchange but is likely caused by a steep increase in external climate forcing. This implies a large change in a major short-lived climate forcing agent, almost certainly tropospheric aerosols. This conclusion is consistent with satellite-based observations of a major decrease in global $SO_2$ emissions from large anthropogenic sources, particularly coal-fired power plants.

There are two potential routes to controlling further rapid temperature rise while health-harming $SO_2$ pollution continues to be

phased out. They are not mutually exclusive, indeed an effective mitigation response with the least possible adverse impacts likely requires both. In the first route, the historical cooling effect of polluting tropospheric aerosols, most of which reside in the planetary boundary layer (PBL) for a relatively short time before precipitating out, is approximately replicated by injecting a smaller quantity of $SO_2$ into the stable troposphere (above the PBL). This geoengineering approach has been criticised for the potential climate shifts it may cause (Trisos et al., 2018). However, humanity has already been engaged for many decades

in major inadvertent geoengineering through its increasing injection of cooling aerosols into the lower troposphere. The approach may not be excessively costly if flue stacks are used which exceed the local PBL height or are located in regions of strong vertical circulation.





The second route involves steep reductions in short-lived and intermediate greenhouse gases, which could partly offset the loss of aerosol cooling. By far the most important of these is methane, with a further contribution from ozone (Hansen et al., 2017). This presents a huge challenge to current energy and food production systems, as the mole fractions of atmospheric methane and tropospheric ozone are currently on the rise as a result of increasing emissions from oil and natural gas systems (Helmig, 2016), (Worden, 2017), with an additional contribution to rising methane from biogenic processes (Worden, 2017). A rapid reversal of this trend, together with the potential involvement of $SO_2$ based geoengineering and intensified adaptation efforts, will be critical for global stability in the next 1–2 decades while the world tackles the still larger problem of $CO_2$ emissions.

**Appendices**

**Appendix 1: Time-series data and forcings**

All time-series data used in the analysis for the paper are monthly values, unless otherwise stated. Global mean stratospheric optical depths used to infer direct radiative forcing by stratospheric aerosols are from NASA (2018) with the exception of values after 2007, where the GISS values are adjusted upwards in response to the recent GloSSAC aerosol analysis (Thomason et al., 2018). Solar direct radiative forcing is obtained by linear interpolation between the annual solar forcing estimates of Sato (see Supplementary Material), which agree closely with monthly estimates of direct radiative forcing based on the NOAA Solar Irradiance Climate Data Record (NOAA, 2018a). Effective radiative forcing, both by aerosols and solar irradiance variations, is calculated as 0.55 × direct radiative forcing, based on climate simulations and satellite measurements of direct short wave and secondary long wave forcing following the Mt. Pinatubo eruption (Hansen, 2005). Based on the extracted effective forcing per unit optical depth τ, 21 W m$^{-2}$/τ (see Supplementary Information), the forcing correction for volcanoes prior to the 1850 − 2017 period modelled by Hansen et al. (2017) is scaled from the value used therein to 0.27 W m$^{-2}$. ENSO-corrected lower troposphere temperature data are taken from Soden et al. (2010). Global mean surface temperature values are taken from the 12-month running mean NASA GISTEMP dataset maintained at Columbia University (2018). Ocean Niño index (ONI) monthly values used to correct the GISTEMP data for ENSO variations in equation (1) of the paper are from NOAA (2018). For consistency with the GISTEMP data, the ONI monthly time series is transformed here to a 12-month running mean. In order to maintain consistency between model curves and measurement data, time series for global temperature forcing are used without smoothing when compared to tropospheric temperature data and smoothed using a 12-month running mean when compared to the smoothed global surface temperature data.

**Appendix 2: Climate response functions**

Response functions for lower troposphere temperature, $P_{LT}$, and global mean surface temperature, $P_{GS}$, are constructed based on the 'intermediate climate response' function of Hansen et al. (Hansen 2007). Both functions are digitised as monthly values and convoluted with effective global temperature forcing, $F_e$, to generate the resultant temperature evolution:



$$T(t) = \lambda \sum P(t') F_e (t - t') \Delta t'$$

where the sum runs over monthly time points (thus $\Delta t = \frac{1}{12}$ y) from January 1850 to the present. The formulation here is pulse

response, formally equivalent to the Greens function approach (Eq. (1) used by Hansen et al. (2017) but arguably more transparent in the context of transient volcano forcing because the pulse response shape mirrors the curve of cooling and temperature recovery after a volcanic eruption. In this formulation the intermediate climate response function used by Hansen et al. (2017) becomes

$$P_H(t) = a_0 \qquad (0 < t \le t_1)$$
$$P_H(t) = a_n/t \quad (t_n < t < t_{n+1})$$

where $t_1 = 1$ y, $t_2 = 10$ y, $t_3 = 100$ y, $t_4 = 2000$ y, and $a_0 = 0.15$, $a_1 = 0.1737$, $a_2 = 008685$, and $a_3 = 0.08345$.


In the above monthly digitized representation, the first year's response has been distributed evenly throughout that year, giving a slightly different early response from that of Hansen et al. (2017). The two responses converge rapidly within a few years, but the distinction is significant during the peak portion of the volcano response: if parameters are correct the present approach is more accurate. In all other respects, the treatments are identical.


The response function using the parameters from Hansen (2017) leads to a transient temperature evolution after volcanic eruptions which is intermediate between the observed lower troposphere response (Fig. 2(a) of the paper) and the global mean surface response ((Fig. 2(b) of the paper). In order to fit more accurately the lower troposphere and global surface responses to the temperature observations the function is empirically modified by varying the time $t_1$, which controls the short-time

behaviour of the response, while keeping the curve continuous at time $t_1$ by adjusting the value of $a_0$. The values of $a_2$ and $a_3$ are then slightly adjusted, by a common factor $\sim 1$, to preserve the normalisation $\sum P(t) \Delta t = 1$. The revised values are reported in the Supplementary Material.

**Competing interests**

The author declares no competing interests.



## Acknowledgement

I thank Lijing Cheng for providing recent data and analysis updates on ocean heat content.

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
