# Peer review of "Current rapid global temperature rise linked to falling SO₂ emissions"

_Earth System Dynamics, 2018_

## Referee Comment (RC1) · Anonymous Referee #1 · 23 Dec 2018

**Review of 'Current rapid global temperature rise linked to falling SO₂ emissions' by Nick E. B. Cowern**

*Summary*

This submitted discussion paper contends that the recent rapid warming in global mean surface temperatures (more commonly referred to as GMST) arises primarily as a result of recent clean air efforts that have reduced the sulphate emissions and thus aerosol burden. That rapid efforts to clean up aerosol emissions could lead to a short-term warming spike has long been recognised. Observational proof of this signal emerging, if indeed it is, would be a valuable addition to the literature. However, as written and analysed I do not find the paper convincing for reasons that I will further articulate below. At its most basic I believe the problem to be too simplified to be able to conclude with requisite confidence a cause. My view is that a substantive rewrite would be required to address these concerns and that these go far beyond major corrections, requiring very extensive work to address.

However, equally, I would not wish my comments to discourage the author from pursuing the work as I think the issue is very important to assess. That said, It is important to recognise that there exists a broad spectrum of opinions within the scientific community as to whether there is value in the analysis of very recent trends and their causes. This is most evident in the analysis of the early 21st Century reduction in warming variously termed a 'hiatus', 'Pause' etc. Personally, I am of the view that there is indeed value in this. If the community do not make efforts to understand recent / ongoing changes I feel that we do global society a disservice. It is natural for decision makers, policy makers and society to want to better understand recent climate and what that portends upon various short to medium term planning horizons.

*Stylistic concerns*

Before going into scientific details, I make a couple of comments on style:

Firstly, I suspect that this had been written for submission to a high profile short-form letters journal and has been resubmitted for publication here following rejection. This is obvious from at least one allusion to a numerical reference as well as the fact that all methods are supplementary information. ESD is not a letters form journal and I find it inappropriate to relegate the methodological detail to the SI. This detail should arguably be front and centre in the paper for this journal. Ultimately this decision, rightly, would rest with the editor, but personally I find the relegation of the necessary methodological detail to the SI unsatisfactory.

Secondly, the piece is grossly under-referenced with references being almost entirely to the work of a single lead author. The lack of citing a balanced sample of the literature around the hiatus and surge is a major issue. Readers need to be correctly oriented to where the piece falls into place holistically within the wealth of recent literature on the topic. At a minimum references to one or more hiatus synthesis / review papers would be appropriate along with associated interpretation thereof in the context of the present piece. Furthermore, there is a lack of citation of data sources. Even after several careful reads I am

none the wiser which surface temperature dataset is being analysed and similarly which OHC, forcing estimates etc. etc.. These are basic issues that greatly negative impact whether the piece is publishable as it stands.

*Analysing at the global mean scale*

The analysis is performed exclusively at the global mean scale. This significantly limits the ability to discriminate between competing hypotheses to the point where, in my judgement, it becomes impossible to unambiguously disentangle causes.

For the past two decades detection and attribution approaches have been searching using correlation or regression approaches for spatio-temporal agreement between model estimates of the responses to forcings and observations. Given the short lifetime of sulphate aerosols and the apparent strong spatio-temporal signal in emissions a properly designed set of model runs could be compared to the observations under e.g. the approach of Allen and Stott, 2000. This would enable a more certain conclusion to be reached. Limiting the analysis to global mean and timeseries congruence is insufficient to alight upon a single principal cause, that is if there is a single principal cause, of observed behaviour. There may be relevant model runs that can be used arising from CPDN citizen science ensemble, the NorESM ensemble described in Outten et al., 2015 or the recently completed very large ensemble at MPI. The single forcings runs of NASA GISS may also be informative.

Failing recourse to a formal detection and attribution analysis, there would be significant value in at least showing and analysing the spatio-temporal evolution in the observations with an associated analysis on plausible aerosol impacts. This may involve recourse to observations of both aerosol and temperature changes resolved spatially. A spatio-temporal correlation of the two fields may be informative in building confidence as to cause and effect.

*Observational uncertainty*

Beyond the fact that I cannot discern which observational record of GMST is employed at present, a much more substantive issue is that only one observational record is analysed. There are numerous surface temperature analyses, including several modern reanalysis datasets (see Simmons et al., 2017 in QJRMS). A list of plausible datasets to consider would include, but not necessarily be limited to:
- HadCRUT4
- Cowtan and Way
- NASA GISS
- NOAA Globtemp
- Berkeley Earth
- JMA
- JRA-55
- ERA-Interim / ERA5

Inclusion of the range of datasets would enable exploration of sensitivity of results to choice of datasets. Furthermore, some of these products are now ensemble products that permit an exploration of parametric uncertainty in these products. Showing that the analysis is

robust to choice of dataset and published dataset uncertainty estimates would greatly improve the analysis.

Similarly, there is uncertainty in estimates of ENSO, OHC and the various forcings being considered (not least the anthropogenic aerosols) and similar consideration of the ensemble of opportunity presented by observationally based estimates, including atmospheric composition reanalysis products, would be greatly beneficial.

I see insufficiently robust analysis of the quality of the OMI aerosol product upon which the attribution inference principally rests. Where is reference to the papers describing the product and its verification? Des the product come with uncertainty estimates? If so, these should be used. What does the spatially sparse AERONET network show? What do available lidars show? What do other space-based instruments capable of discerning aerosol properties such as the hyperspectral sounders show?

*Exclusion of plausible natural modes of variability*

The analysis attempts, reasonably, to remove the impacts of ENSO. That ENSO has a substantive effect on the GMST on inter-annual timescales is well known and beyond dispute, as is the lag (although a reference to support this would be advisable). That said, the use of an apparent multiplicative effect of 0.1 seems unduly deterministic. The available finite sample of ENSO events probably means that only a range, likely centred around 0.1, is defensible. This range should be quantified and used.

The bigger issue is the implicit assumption that once ENSO is removed there exist no other important mechanisms of natural variability. That assumption is, of course, over-simplistic. There are very many major modes of variability that have power across a broad range of timescales and project strongly onto regional and / or global surface temperatures. These modes do not necessarily solely arise in OHC but may be driven by e.g. sea-ice changes or land cover responses. Variability can lead to multi-annual excursions around a long-term trend driven by changes in large-scale climate forcings. Climate model control runs highlight that, as simulated, the climate can support multi-annual to multi-decadal excursions from climatology in the absence of forcings. Such excursions do not result solely from ENSO variability or changes in near-surface ocean layers.

*Earth System Temperature Metric*

I find this metric intuitively interesting. However, the explanation as given is insufficient for me, and therefore presumably your readers, to assess its efficacy or properties. It should be better explained and thought given as to how to prove its utility which may include, for example, its application to various climate model simulations, prior to application to and analysis against real-world observations. This would strengthen the analysis considerably.

*Further questions*

1. The QBO is a stratospheric mode of variability how can it be removed with the earth system temperature metric? The metric does not include any stratospheric

contribution. I find the analysis in this regard unconvincing. It may be ameliorated by moving SI to the main text.

2. There is a question that bedevilled hiatus papers around statistical / practical significance. For short timescales in ARMA series statistical significance even on a decadal scale of changes in rate are highly questionable. I am unconvinced of statistical significance. Then there is the question of practical significance. The hiatus was not statistically significant, but it was for practical intents a departure. I suspect the same ambiguity pertains to the recent surge-like behaviour, at least in so far as it exists to date. Likely several more years of rapid warming would be required to attain statistical significance? Careful thought would appear warranted around how to communicate this issue.

---

## Referee Comment (RC2) · Anonymous Referee #2 · 14 Jan 2019

This manuscript presents an analysis of global mean surface temperature trends, and identifies major changes in climate forcing during the last 1-2 decades. In particular the paper finds a >50% decrease in $SO_2$ emissions from large sources during the last 6 years has reduced tropospheric aerosol cooling and thereby caused an acceleration of anthropogenic global warming. Furthermore the paper finds ocean-atmosphere heat exchange does not contribute substantially to 21st century warming trends, once trends are corrected for ENSO variations.

The article addresses a very important topic, identifying the extent to which tropospheric aerosol forcing, volcanic aerosol and internal climate variability are drivers of observed climate variability in recent decades. However, several of the findings from the paper are already established, for example it is well known that major changes in

climate forcing have occurred in the last 1-2 decades, this is not a new finding. The paper does not, in its current form, put the research into sufficient context with regard to other research to attribution the drivers of recent climate change. In particular, the Introduction section is very weak, only 1 paragraph (lines 24 to 28) explaining the context of the research with no citations of previous research in this area. The rest of the Introduction explains this paper, with only very brief and vague mention of other literature findings.

Overall I was struck that the paper read more like a draft of a graduate student dissertation rather than a paper for peer-reviewed journal and the manuscript requires much more work to explain the methods more clearly, and what additional information they bring for example compared to other similar studies. In particular compared to findings from detection and attribution studies to fingerprint the climate responses to different forcings from climate model integrations (e.g. Hegerl & Zwierz, 2011), as in the current community activity DAMIP (Gillett et al., 2016), alligned to CMIP6.

Also, in several places there are incorrect, inaccurate or unsupported statements in the manuscript (see e.g. specific comments 1, 4, 5) which need to be adequately caveated or better qualified with supporting evidence/reference. In particular, the context of this paper within findings in chapter 10 of the IPCC AR5 climate assessment report (Bindoff et al., 2013) need to be much better explained.

For the above reasons, the paper requires fundamentally re-writing, I am therefore recommending the paper be rejected and re-drafted before re-submission.

Specific comments ——————

1) Abstract, lines 5-6: This first sentence may be correct for some periods of the historical record, but it is certainly not the case during periods of strong volcanic activity (e.g. Santer et al., 2014). Also, the starting 5 words "It is widely held that.." is an unscientific way to begin an article, it may be potentially OK if those words were replaced by "Outside periods of strong volcanic activity...". Perhaps adding "slowly-varying" or

similar before "changes in external forcing" may potentially also help explain what is being contended here. However, since this directly relates to the topic of the article, and the given timescale here is out to a decade, I would advise to keep this statement more open.

2) Abstract, lines 9-11 – the main evidence supporting the papers findings stems from an analysis to compare ENSO-corrected global mean surface temperature trends with a new temperature metric that corrects for changes in ocean heat content (OHC). But the timescales for this central methodology need to be stated here in the Abstract and the reader given more information re: the "scaled OHC", otherwise they will remain yet to be convinced.

3) Abstract, lines 13-19 – The authors refer to isolating the underlying signal of anthropogenic global warming, but again the timescales are unclear here. Related to this, the apparent slowdown in surface warming is referred to as "from late 1990s to 2011" which has one-year uncertainty at end of the period but multiple-year uncertainty at the start of the period. The language used for these statements needs to be sharpened up substantially and the proposed link between recent SO2 emissions decreases and the link between the more rapid warming in observed global mean surface temperature trends in the last 5 years and a decrease in tropospheric aerosol cooling needs to be more than just the timing of SO2 emissions. Reducing SO2 emissions does not necessarily mean less tropospheric aerosol cooling, for example due to ammonium nitrate aerosol forming more effectively as SO2 emissions decrease (e.g. Hauglustaine et al., 2014), and the influential role of anthropogenic organic aerosol (e.g. Tsigaridis and Kanakidou, 2018).

4) Introduction, page 2, line 1 – what is meant by a measure of temperature being "potentially dependent on non-ENSO climate variability"? The sentence does not seem to make sense. Also, the authors contend the ENSO-corrected global mean surface temperature record is then independent of all internal climate variability. The paper needs to explain other sources of inter-annual climate variability such as the North

Atlantic Oscillation, and to also explain the role of decadal internal variability in the Atlantic (the Atlantic Meridional Oscillation, e.g. Sutton and Hodson, 2005) and in the Pacific (the Pacific Multidecadal Oscillation, e.g. Meehl et al., 2008).

5) Section 2.1, page 3, lines 16-18 – the wording is extremely vague, such the reader is not clear which time-period is ebing discussed here. Furthermore the statement "decadal variations caused by changes in climate forcing have been thought to be small". This statement is so clearly false, since greenhouse gas forcing, with offset from cooling from increased tropospheric aerosol forcing is very well established in successive climate assessment reports to be the primary drivers of climate change in the last 5 decades.

References ————-

Bindoff, N. L., Stott, P. A., AchutaRoa, K. M. et al. (2013) Detection and Attribution of Climate Change: from Global to Regional, Chapter 10 of "Climate Change 2013: the Physical Science Basis. Contribution of Working Group 1 to the Fifth Assessment Report of the Intergovernmental Panel of Climate Change", Cambridge University Press, 2013.

Gillett, N. P., Shiogama, H., Funke, B., Hegerl, G. et al. (2016) The Detection and Attribution Model Intercomparison Project (DAMIP v1.0) contribution to CMIP6, Geosci. Model Dev., 9, 3685–3697, 2016.

Hauglustaine, D. A., Balkanski, Y. and Schulz, M. (2014) A global model simulation of present and future nitrate aerosols and their direct radiative forcing of climate, Atmos. Chem. Phys., 14, 11031–11063, 2014.

Hegerl, G. and Zwiers F. (2011) Use of models in detection and attribution of climate change WIRES Climate Change Reviews, vol. 2, pp. 570-591, 2011.

Meehl et al. (2008) The Mid-1970s Climate Shift in the Pacific and the Relative Roles of Forced versus Inherent Decadal Variability J. Climate, vol. 22, pp. 780-792, 2008.

Santer, B. D., Bonfils, C., Painter, J. F. et al. (2014) Volcanic contribution to decadal changes in tropospheric temperature Nature Geoscience, vol. 7, pp. 185-189, 2014.

Sutton, R. and Hodson, D. (2005), Atlantic Ocean Forcing of North American and European Summer Climate Science, vol. 309, pp. 115-118, 2005.

Tsigaridis, K. and Kanakidou, M. (2018) The Present and Future of Secondary Organic Aerosol Direct Forcing on Climate Current Climate Change Reports, 4, pp. 84–98, 2018.
* * *